# Role of Oxidative Stress in the Senescence Pattern of Auditory Cells in Age-Related Hearing Loss

**DOI:** 10.3390/antiox10091497

**Published:** 2021-09-21

**Authors:** Luz del Mar Rivas-Chacón, Sofía Martínez-Rodríguez, Raquel Madrid-García, Joaquín Yanes-Díaz, Juan Ignacio Riestra-Ayora, Ricardo Sanz-Fernández, Carolina Sánchez-Rodríguez

**Affiliations:** 1Department Clinical Analysis, Hospital Universitario de Getafe, Carretera de Toledo, km 12.500, Getafe, 28905 Madrid, Spain; luzmar.rivas@salud.madrid.org; 2Department of Medicine, Faculty of Biomedical and Health Sciences, Universidad Europea de Madrid, Villaviciosa de Odón, 28670 Madrid, Spain; sofia.mrr@hotmail.com (S.M.-R.); raquel.madrid@universidadeuropea.es (R.M.-G.); juanignacio.riestra@salud.madrid.org (J.I.R.-A.); 3Department Otolaryngology, Hospital Universitario de Getafe, Carretera de Toledo, km 12.500, Getafe, 28905 Madrid, Spain; joaquin.yanes@salud.madrid.org (J.Y.-D.); rsanzf@salud.madrid.org (R.S.-F.)

**Keywords:** age-related hearing loss, senescence, auditory cell, hydrogen peroxide (H_2_O_2_), oxidative stress

## Abstract

Age-related hearing loss (ARHL) is an increasing and gradual sensorineural hearing dysfunction. Oxidative stress is an essential factor in developing ARHL; additionally, premature senescence of auditory cells induced by oxidative stress can produce hearing loss. Hydrogen peroxide (H_2_O_2_) represents a method commonly used to generate cellular senescence in vitro. The objective of the present paper is to study H_2_O_2_-induced senescence patterns in three auditory cell lines (House Ear Institute-Organ of Corti 1, HEI-OC1; organ of Corti, OC-k3, and stria vascularis, SV-k1 cells) to elucidate the intrinsic mechanisms responsible for ARHL. The auditory cells were exposed to H_2_O_2_ at different concentrations and times. The results obtained show different responses of the hearing cells concerning cell growth, β-galactosidase activity, morphological changes, mitochondrial activation, levels of oxidative stress, and other markers of cell damage (Forkhead box O3a, FoxO3a, and 8-oxoguanine, 8-oxoG). Comparison between the responses of these auditory cells to H_2_O_2_ is a helpful method to evaluate the molecular mechanisms responsible for these auditory cells’ senescence. Furthermore, this in vitro model could help develop anti-senescent therapeutic strategies for the treatment of AHRL.

## 1. Introduction

Age-related hearing loss (ARHL), or presbycusis, is a pathology that affects 33% of the population worldwide [1]. ARHL is observed simultaneously in both ears in the high frequency spectrum and is a multi-factorial process involving complex interactions between intrinsic (genetic) and extrinsic (environmental) factors [1,2]. ARHL is characterized by cochlear dysfunction, which includes loss of sensory cells, atrophy of the stria vascularis, and loss of the spiral ganglion neurons [1,2]. Establishing the intrinsic mechanisms responsible for the disease will drive the development of preventive therapies [1]. The mechanisms that have been described are: imbalance between the production of radical oxygen species (ROS) and antioxidants, cumulative mitochondrial dysfunction, accumulation of mitochondrial DNA damage, excitotoxicity, and cellular senescence leading to apoptotic and necrotic auditory cell death [3,4]. However, the exact biological mechanism of ARHL is still unknown.

Senescence is considered a basic mechanism of aging [5]. Cellular senescence limits the proliferation of damaged or aged cells [6,7]. Although senescence plays an essential role in tissue homeostasis and physiology in normal development, senescence is a response to stress associated with aging, such as telomere attrition and genomic alterations, which are aging characteristics. Cellular senescence can be divided into several types, such as replicative senescence resulted from telomere shortening [8], stress-induced premature senescence (SIPS), which is caused by an oncogene [9], or a variety of stressors (chemotherapeutic agents or oxidative stress) [10].

Numerous aging pathologies associated with senescence have been studied, as in glaucoma [11], cataracts [12], the diabetic pancreas [13], osteoarthritis [14], and cancer [15]. In age-related disorders, senescent cells accumulate in tissues associated with the aging process through the senescence-associated secretory phenotype (SASP). As the disease advances, an additional surge of senescent cells is produced at the damaged sites [16]. In addition, senescent cells induce high production of mitochondrial-derived ROS [15]. The SASP and enhanced ROS production contribute to cellular senescence in both autocrine [17,18] and paracrine ways [19,20].

In recent years, different studies have confirmed that senescent cells act on their environment through mitochondrial dysfunction. Mitochondrial dysfunction is considered a further “hallmark of aging” [21], and, together with cellular senescence, both have been independently recognized as important leads of aging [22,23,24,25]. Furthermore, cellular senescence promotes senescence-associated mitochondrial dysfunction (SAMD) [17]. Cumulative evidence indicates that mitochondria in senescent cells exhibit various changes in mass, function, and structure. In general, the number of mitochondria increases in senescent cells [17,26,27]. However, although mitochondria are more abundant, they are dysfunctional. In senescent cells, mitochondria show increased ROS production [17,25].

Senescence of different cells is the most used in vitro experimental model for cellular aging [28]. The induction of oxidative stress is often used for studies based on SIPS. To date, distinct inducers of oxidative stresses have been utilized to induce SIPS in vitro, such as hydrogen peroxide (H_2_O_2_) [29], ultra-violet light [30], tert-butyl hydroperoxide [31], and ethanol [31]. H_2_O_2_ is the foremost ordinarily utilized inducer of senescence because it is a natural inducer of oxidative stress [28]. Oxidative stress can induce premature senescence in auditory cells, resulting in ARHL. Understanding auditory cell behavior under oxidative stress would be important to study how to prevent, reverse or postpone senescence induced by oxidative stress in auditory cells cultures.

In recent years, different approaches have been used to establish the molecular mechanisms of ARHL, but these are often limited by the low amount of tissue available for assays, the difficulty of obtaining organotypic cultures along with the need to perform multiple explants for each experimental condition or the few animal models available [32]. One solution to the problem would be the culture of auditory cells, but this is complicated because the mammalian organ of Corti cells do not proliferate. Currently, this has been resolved by the obtaining of several auditory cell lines from transgenic mice (Immortomouse1) as HEI-OC1 (House Ear Institute-Organ of Corti 1), OC-k3 (from organ of Corti), and SV-k1 (from stria vascularis) cells [33,34,35], which proliferate indefinitely.

The data of this study showed that H_2_O_2_ induced senescence in HEI-OC1, OC-k3, and SV-k1 auditory cells. Different lines of evidence obtained in this research support these results. Therefore, this model of senescence induced by oxidative stress in the cell lines HEI-OC1, OC-k3, and SV-k1 can be a useful tool to study the intrinsic mechanisms responsible for ARHL and allow the development of anti-senescence therapies and prevent ARHL.

## 2. Materials and Methods

A comprehensive overview of the experimental timeline and procedures that have been used in this study is described in Figure 1.

### 2.1. Cell Culture and H_2_O_2_ Treatment

The three mouse auditory cell lines, SV-k1, HEI-OC1 and OC-k3, were kindly provided by: Dr. F. Kalinec (House Research Institute, Los Angeles, CA, USA), Dr. Maria Rosa Aguilar (Department of Nanomaterials and Polymeric Biomaterials Institute of Science and Technology of Polymers CSIC, Madrid, Spain), and Dr. Beatriz Duran Alonso (Institute of Molecular Biology and Genetics (IBGM), University of Valladolid, Valladolid, Spain), respectively. The cells were cultured in Dulbecco’s high glucose Eagle’s medium (DMEM; Gibco BRL, Waltham, MA USA) with 10% fetal bovine serum (FBS; Gibco BR, Waltham, MA USA), without antibiotics at 33 °C and 10% CO_2_, as previously described [36].

Cells were treated with H_2_O_2_ (VWR Inc., West Chester, PA USA) at different concentrations (50, 100, 150, 200 or 400 μM; 1, 2 or 5 mM) for 1 or 2 h and post-treatment analysis was performed at 24, 48, and 72 h. Cell viability assays were performed under these experimental conditions. In the following experiment settings, H_2_O_2_ concentrations, the treatment times (1 and 2 h) and post-treatment analysis (24, 48, and 72 h) varied depending on the experiments.

### 2.2. Cell Viability Assay

HEI-OC1, OC-k3, and SV-k1 were seeded on 96-well plates at 2 × 10^4^ cells/mL. After 24 h, cells were treated with eight H_2_O_2_ concentrations for 1 or 2 h. Then, a new medium was added until cell viability assay was performed. At 24, 48, or 72 h post-treatment, PrestoBlue cell viability assay (Invitrogen, Waltham, MA USA) was performed following the manufacturer’s protocol. Fluorescence was measured with a FLUOstar Omega (BMG Labtech, Ortenberg, Baden-Württemberg, Germany) plate reader. In the control cells group, the average OD was taken as 100% of viability.

### 2.3. Analysis of the Morphological Characteristics of Cells

Cells were seeded in 24-well plates at 5 × 10^4^ cells/well and treated as described above. The study of the morphological characteristics of cells was performed under a bright field with the Olympus CKX41 microscope and the coupled cellSens Entry imaging system.

### 2.4. Capacity of Cells to Reseeding Post H_2_O_2_ Treatments

After treatment with H_2_O_2_ (1 and 2 h), the adhesion cell capacity was verified by replating in 24-well plates upon reaching 80% confluence. H_2_O_2_-treated cells were seeded at a 1:1 ratio, and untreated cells were seeded at a 1:3 ratio. Cells were observed on successive days to check for complete growth arrest and cell proliferation capacity.

### 2.5. Senescence-Associated β-Galactosidase Assay

Cells were seeded in the Lab-Tek II Chamber Slide System of 8-well plates (Nunc) at 3 × 10^4^ cells/well. After 24 h, cells were treated with H_2_O_2_ (50, 100, 150, 200 and 400 μM) for 1h. To evaluate senescence, the Galactosidase Detection Kit (Abcam, Cambridge, UK) was used after the end of H_2_O_2_ treatment (at 24, 48, or 72 h) according to the fabricator’s instructions. The number of not colored (negatives) and blue (positive, biomarker of cellular senescence) cells were counted in a minimum of three random fields in each sample. Cells were visualized in an Olympus BX51 microscopy.

### 2.6. Determination of Population Doubling Rate

The cell population doubling rate was determined by the method described above by Young et al., 2010 [37]. Cells were cultured in 6-well plates at 1 × 10^5^ cells/well; after 24 h, cells were treated 1 h with H_2_O_2_, replaced with a standard culture medium, and then incubated. Then, 1 × 10^5^ cells/well were reseeding for the following recount, every 72 h, up to 3 passages. The population doubling rate was calculated using the formula: [log (number of cells counted/well)−log (number of cells seeding)]/log 2.

### 2.7. Mitochondrial Superoxide Anion Detection

We used MitoSOX Red fluorochrome (Invitrogen, Waltham, MA USA) to determine the superoxide anions produced in mitochondrial. Cell lines were plated onto a Lab-Tek II Chamber Slide System of 8-well plates at 2 × 10^4^ cells per well. After 24 h in standard conditions, cells were treated with different concentrations of H_2_O_2_ for 1 h. MitoSOX assay was performed at 24 h post-H_2_O_2_ treatment. Then, cells were washed and incubated with 2.5 μM MitoSOX-Red reagent for 45 min at 33 °C in the dark. Cell’s fluorescence was visualized by Olympus BX51 microscope.

The cells were then stained with 200 nM of Tracker Green staining kit (Abcam, Cambridge, UK) for 45 min at 33 °C in the dark to corroborate the mitochondrial localization of MitoSOX-Red. Once active mitochondria are labeled with MitoTracker, they can be detected by fluorescence microscopy. The fluorescence intensities of MitoTracker-Green, MitoSOX-Red, and merge were quantified using the Image J program.

Briefly, the level of fluorescence in a given region (e.g., nucleus or cytoplasm) was determined by the ImageJ program as follows: (1) the cell of interest was selected using the freeform drawing/selection tools; (2) the area, integrated density and mean gray value were selected; (3) from the Analyze menu “set measurements” was selected; (4) “measure” was then selected from the analyze menu, prompting a pop-up box to appear with the values for that first cell; (5) then, a region next to the cell that has no fluorescence was selected to serve as the background; (6) this step was repeated for the other cells in the field of view of measurement; (7) all data in the results window were selected, and copied and pasted into a new Excel worksheet; (8) finally, the following formula was used to calculate the corrected total cell fluorescence (CTCF): CTCF = integrated density—(area of selected cell * mean fluorescence of background readings).

### 2.8. Reactive Oxygen Species Detection

To measure cytosolic superoxide generation, red fluorescent probe dihydroethidium (DHE; Calbiochem, San Diego, CA USA) was used [38]. However, DHE can also detect ONOO^−^ or ^•^OH [38,39]. Briefly, cells were seeded in the Lab-Tek II Chamber Slide System of 8-well plates at 2 × 10^4^ cell/well. After 24 h, cells were treated with different H_2_O_2_ concentrations and incubated for 24 h. Then, after fixing the cells with 4% paraformaldehyde (PFA) for 10 min, they were incubated with probe DHE (4 µmol/L) at 37 °C for 90 min in the dark. After, cells were stained with 300 nM of 4′,6-diamidino-2-phenylindole dihydrochloride (DAPI) (Sigma-Aldrich, San Luis, AZ USA) at 37 °C for 5 min in the dark. DAPI is a reagent that emits blue fluorescence marking the cell nucleus. Cells were observed on an Olympus BX51 microscope, and the ImageJ program analyzed the images as described above.

### 2.9. Indirect Inmmunofluorescence

After treatment with H_2_O_2_, the cells were fixed with 4% PFA for 10 min and blocked at 37 °C for 1 h. Then, cells were incubated with the antibodies Forkhead box O3a (FoxO3a) (1/200) and 8-oxoguanine (8-oxoG) (1/50) (Abcam, Cambridge, UK) overnight at 4 °C. Following this, the secondary antibody Alexa Fluor 546 (1/250; Molecular Probes, Eugene, OR, USA) was added to the cells at 37 °C for 45 min. Subsequently, the cells were stained with DAPI (Sigma-Aldrich, San Luis, CA, USA) at 37 °C for 5 min. Cells were observed on an Olympus BX51 microscope, and the ImageJ program analyzed the images as described above. The suppression of the primary antibody evaluated the specificity.

### 2.10. Statistical Analysis

Statistical values were calculated by variance ANOVA test and Tukey’s multiple comparison test using the SPSS 19.0 software. Data are shown as mean ± standard deviation (SD). *p* < 0.05 values indicate statistical significance.

## 3. Results

### 3.1. Preliminary Screening of H_2_O_2_ Treated Cells: Cell Viability Analysis

Cell viability results underline that HEI-OC1, OC-k3, and SV-k1 cells treated with H_2_O_2_ exhibited decreased cell viability compared to untreated cells in a dose and time-dependent manner. In Figure 2A, we show the HEI-OC1 cell viability after 1 h of treatment. H_2_O_2_ reduced cell viability below 70% (cell viability lower than 70% is cytotoxic according to ISO 10993-5: 2009) at 24, 48 and 72 h, and when treatment was equal or higher than 50 μM H_2_O_2_, the differences between untreated and treated cells were significant (*p* < 0.05). Treatment of HEI-OC1 for 2 h with H_2_O_2_ decreased cell viability more markedly, with respect to 1 h of treatment for concentrations of 50 μM onwards. In particular, the decrease in cell viability is more pronounced at 72 h (*p* < 0.05) in cells treated with high concentrations of H_2_O_2_ (Figure 2A).

As for OC-k3, we analyzed cell viability also at 1 and 2 h of exposure to H_2_O_2_ and at 24, 48 and 72 h post-treatment. In general, at 1 h, we found that a decrease in cell viability below 70% was observed at concentrations greater than 100 μM for 24 and 48 h, while at 72 h, this concentration decreased to 50 μM (Figure 2B). When comparing these results with 2 h of treatment, we verify that cell viability decreases more markedly from lower concentrations, such as 50 μM, at 24, 48 and 72 h post-treatment. It should be noted that, at 72 h post-treatment, the decrease in cell viability is significantly lower, at all concentrations, with respect to 24 and 48 h.

Cell viability data for the SV-k1 cell line are similar to those for the other two cell lines, with viability decreasing as a function of concentration and time. Thus, it is observed that the decrease in viability is more pronounced at 2 h of treatment and at 72 h post-treatment (Figure 2C).

In brief, of the three cell lines, HEI-OC1 cells are the most sensitive to H_2_O_2_ treatment, with a significant decrease in cell viability at lower concentrations (50 μM) compared to OC-k3 and SV-k1 cells.

Since we obtained a very marked decrease in cell viability with exposure to H_2_O_2_ for 2 h, we choose the 1 h condition for additional assays.

### 3.2. Morphological Changes and Capacity of Cells to Reseeding Post H_2_O_2_ Treatments

The three cell lines studied post-H_2_O_2_ administration revealed changes in their morphology as a function of the increase in H_2_O_2_ concentrations. In fact, morphological changes were found from the smallest concentration (50 μM), with these changes being more pronounced at the highest concentrations (200, 400 μM; 1, 2 and 5 mM) at 1 h (Figure 3). The changes were also more pronounced depending on the time elapsed after the end of the treatment (24, 48 and 72 h). In general, cells exhibited marked morphological changes observed to range from thinner and longer cells to enlarged and ramified cells and alterations in the organelles, all characteristics of senescent cells [40,41] (Figure 3). Data for 24 and 48 h after H_2_O_2_ treatment are not shown. It should be noted that following the results of cell viability (Figure 3), the number of cells in all types significantly decreases concerning their respective controls as a function of time and H_2_O_2_ concentration. Additionally, increasing the exposure time and H_2_O_2_ concentration slowed down auditory cell growth after 2 and 5 mM H_2_O_2_ treatment, and the cells were not viable on re-seeding. In contrast, cells that had been treated with 50,100, 150, 200, 400 μM, and 1 mM H_2_O_2_ were able to undergo a minimum of 3 passages.

### 3.3. H_2_O_2_ Increase SA β-Galactosidase Activity in Cells

The next step was to evaluate the H_2_O_2_ effect on SA β-Gal activity in the three cells lines. In all experimental conditions, the enzyme activity increased with H_2_O_2_ treatment in comparison with the controls (untreated cells). Under these experimental conditions, after a treatment of 1 h, all concentrations of H_2_O_2_ tested effectively increased the activity of β-Gal of SA at 24 and 48 h post-treatment, with a prolonged effect up to 72 h (Figure 4). Figure 4 also shows results from an illustrative experiment and the corresponding cell images of 1 h H_2_O_2_ exposure and 24 h post-treatment (Figure 4A). We determined that the percentage of positive cells (blue cells) after each treatment with H_2_O_2_ was higher than in controls (Figure 4B–D).

Comparing assay results, HEI-OC1, OC-k3, and SV-k1 cells have different ways of responding to oxidative stress (Figure 4B–D). The SA β-Gal activity observed in HEI-OC1 and OC-k3 cells depended on exposure time and H_2_O_2_ concentration, reaching a maximum activity with 2 mM concentration of H_2_O_2_ for 1 h and 72 h post-treatment, while SV-k1 cells showed the highest increase with 2 mM H_2_O_2_ for 1 h and 24 h post-treatment. When the time elapsed at the end of the treatment with H_2_O_2_ is 48 and 72 h, the SV-k1 cells do not present differences concerning the control, no SA β-Gal activity is observed even in high concentrations (Figure 4B–D).

At each experimental time point, 1, 2, and 5 mM H_2_O_2_ produced a cytotoxic effect and, therefore, these high concentrations were not used for the following tests.

### 3.4. Population Doubling Rate Post H_2_O_2_ Treatments

In addition, we determined the cell duplication rate to test the aging model. Lower rates indicate a slower speed of cell growth, indicating cell senescence. A progressive decrease in the doubling rate of the cell population was obtained depending on the concentration of H_2_O_2_. Figure 5A shows that the HEI-OC1 cell line decreased the duplication rate in the first two passages, and in the third, it increased again. Control cells grew normally during all passages, maintaining the doubling rate.

In the OC-k3 and SV-k1 cell lines (Figure 5B,C), the control cells showed a high capacity to duplicate again in all passages. On the contrary, the treated cells progressively decreased their duplication capacity as the H_2_O_2_ concentration increased, decreasing dramatically from the 100 µM concentration at the third passage.

### 3.5. H_2_O_2_-Treatment Induced Mitochondrial ROS in Cells

H_2_O_2_-treated cells were marked with Mitotracker-Green to localize active mitochondria and control for the total amount of mitochondria, and with MitoSOX-Red to check mitochondrial ROS production. The data obtained show significantly increased ROS at 24 h H_2_O_2_ post-treatment compared to untreated conditions in the three cell lines (Figure 6, Figure 7 and Figure 8). Of note, H_2_O_2_ treatment modified the total number of mitochondria (Figure 6, Figure 7 and Figure 8). We found an increase of mitochondria in cells treated with H_2_O_2_ vs. control.

### 3.6. H_2_O_2_ Increase Levels of Reactive Oxygen Species

Fluorescence for an oxidized dihydroethidium (DHE) probe was used to detect ROS by visualizing the increased intensity of red fluorescence in the nuclei of cells. Cells cultured in the H_2_O_2_ groups (Figure 9) showed increased fluorescence compared to the control group. Of the three cell lines, SV-k1 cells showed lower levels of ROS than HEI-OC1 and OC-k3. Statistical significance was obtained by analyzing fluorescence using ImageJ software; thus, for HEI-OC1 cells, H_2_O_2_ showed a DHE intensity of 28.1 ± 0.6 at 400 μg/mL arbitrary units (AU) (*p* < 0.05) vs. the CTR group, which showed 11.0 ± 0.4 AU (Figure 9D).

ROS levels in OC-k3 cells increased to levels of 24.2 ± 0.8 AU at a concentration of 400 μg/mL, as compared to the control, which was 8.5 ± 0.2 AU (Figure 9E). In SV-k1 cells, for the concentration of 400 μg/mL, levels of 13.2 ± 0.3 AU were obtained, while for the control, the value was 7.4 ± 0.06 AU (Figure 9F).

### 3.7. H_2_O_2_ Treatment Increased Levels of 8-Oxoguanine in Cells

Oxidative stress, which causes DNA damage through 8-oxoG formation, was measured in cells at 24 h after 1 h of H_2_O_2_ treatment (Figure 10). We did not obtain differences in nuclear 8-oxoG levels between auditory cells treated and not treated with H_2_O_2_ (Figure 10); however, we visualized that cytoplasmic 8-oxoG immunoreactivity was significantly increased in auditory cells in the presence of H_2_O_2_ (Figure 10). Previous studies have shown that 8-oxoG staining also occurs in the presence of mitochondrial RNA and DNA [42].

At 24 h post-exposure of H_2_O_2_ (1 h), the three lines of auditory cells increased 8-oxoG fluorescence intensity at the maximum concentrations of H_2_O_2_ (Figure 10). In HEI-OC1 and OC-k3 cells, treatment with H_2_O_2_ increased 8-oxoG fluorescence, but only significantly from the 200 μM concentration. In contrast, 8-oxoG staining was significantly increased from 150 μM in SV-k1 cells (Figure 10).

### 3.8. Foxo3 Increased in Cells Post H_2_O_2_ Treatments

Finally, we measured the expression of transcription factor Foxo3, because it is involved in a variety of functions, such as cellular differentiation, control of cell cycle, resistance to apoptosis and oxidative stress, and inhibition of cellular proliferation [43]. The results showed a significant increase in Foxo3 levels for cells after H_2_O_2_ treatment. In fact, we observed significant increases of Foxo3 in the cell lines HEI-OC1 and OC-k3 from the concentration of 150 µM of H_2_O_2_, while in the SV-k1 cells, it was significant from a concentration of 100 µM (Figure 11).

## 4. Discussion

Cellular senescence contributes to age-associated tissue damage as well as age-related pathologies, such as ARHL. The availability of valid and rapid methods to produce senescence in vitro will facilitate the investigation of the mechanisms of senescence, the susceptibility to senescence among different populations of auditory cells, and potential anti-senescence therapeutic targets. To this end, we developed an auditory cell model of senescence by using several concentrations of H_2_O_2_ as an inductor of premature senescence in HEI-OC1, OC-k3 and SV-k1 auditory cells. This study is the first to demonstrate that oxidative stress induced senescence in three types of auditory cell populations. H_2_O_2_ has been utilized to induce senescence in other cell types, such as vascular endothelial cells [44], human adult and perinatal tissue-derived stem cells [45], and HEI-OC1 cells [46,47].

In the present study, we attained exciting findings on the induction of senescence in an in vitro model with auditory cell lines from transgenic mice. These findings are as follows: 1. Cell viability is impaired in auditory cells treated with H_2_O_2_; 2. The population doubling rate decreases in auditory cells treated with H_2_O_2_; 3. β-galactosidase activity increases in auditory cells treated with H_2_O_2_; 4. There is increased mitochondrial mass and ROS overproduction post-H_2_O_2_ treatment; 5. There is an increase of 8-oxoG levels (DNA damage by oxidative stress) post-H_2_O_2_ treatment; and 6. Foxo3 levels increased in auditory cells after H_2_O_2_ treatment. Combining the above features is the best choice for detecting senescent cells, because any single feature can be found outside of senescence or may appear at different times during transformation to a senescent cell state [48,49]. No single marker that unequivocally identifies senescent cells has been identified so far [48,49].

To this end, we demonstrated that H_2_O_2_ decreased cell viability in a dose- and time-dependent manner in all three cell lines but also affected antiproliferative effects at a long time after the end of exposure (up to 24, 48, or 72 h). Our results are coherent with the decrease of cell vitality shown in human adipose tissue-derived stem cells (hASCs) and human Wharton’s jelly-derived MSCs (hWJ-MSCs) treated with concentrations of H_2_O_2_ (up to 400 µM) [45]. The obtained differences in cell viability between HEI-OC1 cells and the other two cell lines may entail the finding confirming different senescence susceptibilities. The same H_2_O_2_ concentrations markedly inhibited proliferation in HEI-OC1 cells and did not significantly affect OC-k3 and SV-k1 cells.

We also found a growth deceleration and an inability to re-seed cells treated with H_2_O_2_ at 2 and 5 mM. These results do not agree with those obtained by Tsuchihashi et al. (2015) and Lin et al. (2019), where HEI-OC1 cells exposed to 1, 2, and 5 mM H_2_O_2_ for 1 h resulted in the absence of a notable decrease in cell viability [46,47]. In our study, for the concentration of 5 mM at 1 h of treatment and 24 h to the end of treatment, cell viability was reduced below 20%. Moreover, cells treated with 50, 100, 150, 200, 400 µM and 1 mM H_2_O_2_ were able to undergo a minimum 3 passages. H_2_O_2_ exposure induced morphology changes in these cells. These results agreed with those obtained by Facchin et al. (2018), in which hASC and hWJ-MSC cells treated with H_2_O_2_ increased in size, with an increase of granules in the cytoplasm and with the ability to withstand up to 3–4 passages post- H_2_O_2_ treatment [45].

We confirmed that the population doubling rate post-H_2_O_2_ treatments in the three cells lines were significantly decreased in our model in a concentration-dependent manner. Lower rates indicate a slower cell growth rate. Our data are supported by previous reports that H_2_O_2_ concentrations (1, 2, and 5 mM) did affect the growth rate of HEI-OC1 cells treated for 1 h [46,47].

These results were confirmed by data obtained from our test of β-Galactosidase activity after 24, 48, and 72 h post-H_2_O_2_ treatment, where HEI-OC1, OC-k3, and SV-k1 cells showed a rise in β-galactosidase activity. In concrete terms, the major senescent effect was evident at 72 h for HEI-OC1 and OC-k3 cells but not for SV-k1 cells. The same trend was found in other research with different cell types, such as HEI-OC1 cells, although these results were obtained at different experimental times [46,47,50,51].

In senescent cells, not only is the cell size increased but the mitochondrial mass is also increased [52]. Previous studies have shown that an increase in mitochondrial accumulation occurs when senescence occurs in different cell types, such as fibroblasts [23], epithelial cells [53], neurons [54], or enterocytes, which develop a senescence phenotype in response to continuous DNA damage [55]. The mitochondria of senescent cells frequently show a reduced membrane potential and, simultaneously, an increased production of ROS [17,56], suggesting dysfunctionality. The present study results were consistent with these concepts, first due to the increase in the mass/number of mitochondria, and second, due to the increase in the generation of mitochondrial and intracellular ROS associated with cellular senescence. ROS occur in numerous forms, including radical hydroxyl radical species (˙OH) and superoxide anions (O_2_^−^), as well as non-radical species, such as H_2_O_2_. These events are associated with the aging process in many tissues and an increase in the number of senescent cells in the same tissues [55,57]. Therefore, during the development of cellular senescence, mitochondrial dysfunction could activate different cellular signaling pathways.

ROS can induce cellular damage by damaging lipids and proteins, as well as attacking DNA directly or indirectly, leading to various pathologies, cancer, and aging [58,59]. One type of this ROS damage is the direct attack on the guanine bases of DNA, resulting in 7,8-dihydro-8-oxoguanine (8-oxoG) [60]. This change has been associated with aging, cancer, and mutagenesis. Our findings highlighted that exposure of cells to H_2_O_2_ induced high 8-oxoG levels, supporting its ability to induce oxidative DNA damage. In agreement with our results, Shen et al. [61] demonstrated that DNA damage caused by 8-oxoG to cochlear hair cells (HC) is essential for developing noise-induced hearing loss (NIHL). In addition, several studies have demonstrated the increase of 8-oxoG levels in age-related pathologies, such as Alzheimer’s, Parkinson’s, and Huntington’s disease [62,63,64]. Specifically, Leon et al. [65] showed that 8-oxoG in mitochondrial DNA, induced by oxidative stress, causes mitochondrial dysfunction and alters neuritogenesis in cultured mouse cortical neurons.

The Forkhead (Foxo) family are key regulators of longevity that promote cell survival through stress repair mechanisms [66,67]. In response to severe stress, Foxo factors can induce cell death in senescent cells, such as apoptosis by the p53 protein [68]. In this study, we found that Foxo3 levels increased predominantly in the cytoplasm of auditory cells after H_2_O_2_ treatment. In agreement with our results, Du et al. [69] demonstrated that circular Foxo3 RNA (circ-Foxo3) was highly expressed in the cytoplasm of cardiac cells from aged patients and mice and induced cellular senescence by modulating multiple stressors and senescence responses. Finally, they also showed that by silencing circ-Foxo3, they inhibited cellular senescence. Other investigations suggest that Foxo3 is related to the ROS of mitochondria in HC of the mouse cochlea. Liu et al. [70] report that neomycin-induced HC damage triggered up-regulation of Foxo3 and production of mitochondrial ROS along with low expression of antioxidant enzymes. β-catenin overexpression in HC inhibited Foxo3 expression and decreased ROS accumulation after injury by neomycin.

## 5. Conclusions

Our findings show, for the first time, the effect of the H_2_O_2_ treatment method in the acquisition over time of the senescence phenotype in three types of auditory cell populations. Identifying the so-called hallmarks of aging has helped develop aging research and focus on delaying different age-related pathologies by targeting the senescence process. In-depth knowledge of the senescence effect in the ARHL will help understand this disease pathogenesis and aid in the development of effective preventive and therapeutic strategies.

In summary, recent years have revealed a crucial role for senescence in the aging process. The development of powerful tools to analyze this relationship should improve our comprehension of the mechanisms through which the growth of senescent cells in organisms leads to age-related diseases. It should also contribute to the development of new therapeutic advances, which could improve the treatment of specific disorders, such as ARHL, and the general health span of aged persons.

## Figures and Tables

**Figure 1 antioxidants-10-01497-f001:**
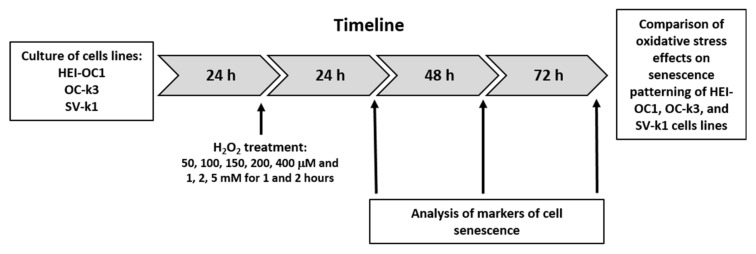
Comprehensive overview of the experimental timeline and procedures.

**Figure 2 antioxidants-10-01497-f002:**
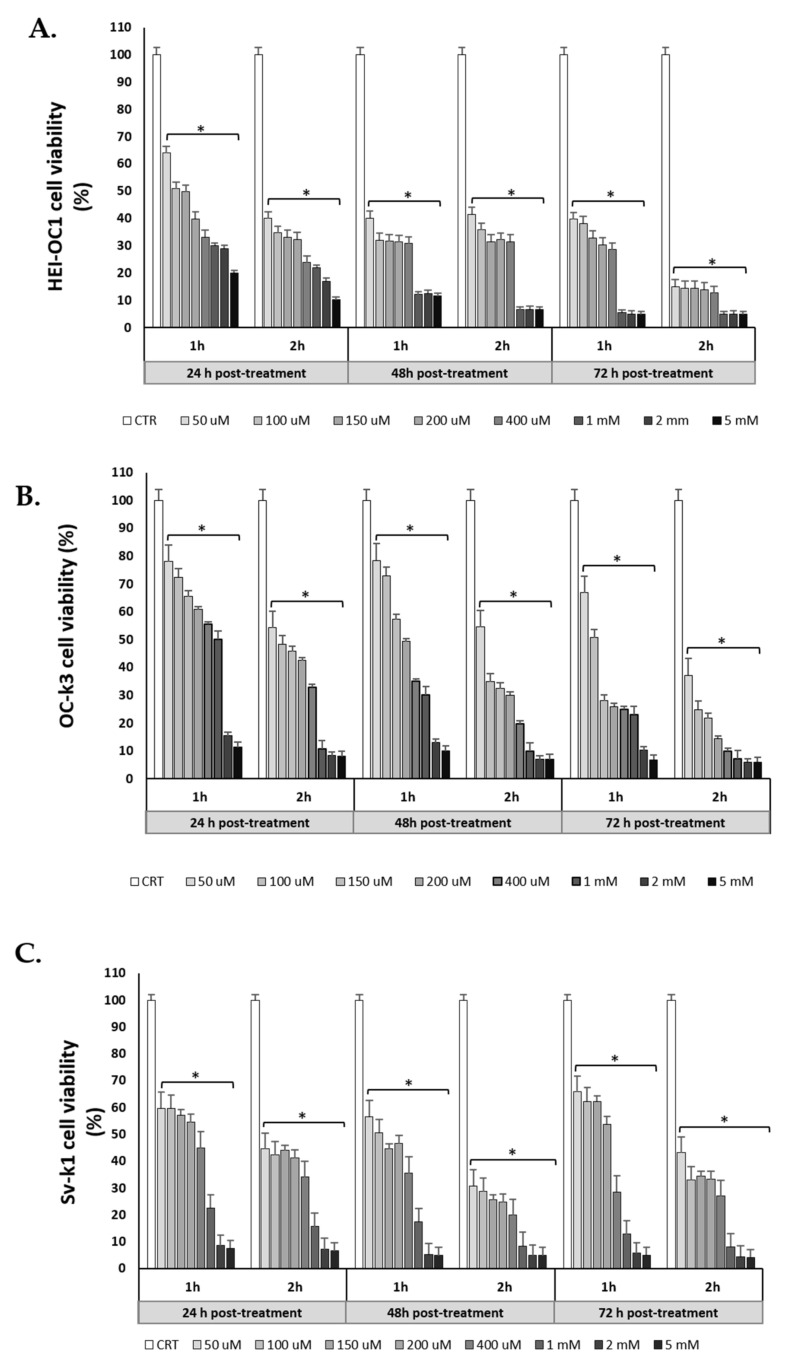
H_2_O_2_ decreases the viability of cells. Experimental conditions: untreated cells are control (CTR) and H_2_O_2_ concentrations are as follows: 50, 100, 150, 200 and 400 μM, and 1, 2 and 5 mM, for 1 and 2 h. Cell viability was measured by PrestoBlue assay in cells after the end of the H_2_O_2_ treatments at 24, 48 and 72 h. (**A**) HEI-OC1 cells; (**B**) OC-k3 cells; and (**C**) SV-k1 cells. Data represent means ± SD (*n* = 3). * *p* < 0.05 vs. CTR group.

**Figure 3 antioxidants-10-01497-f003:**
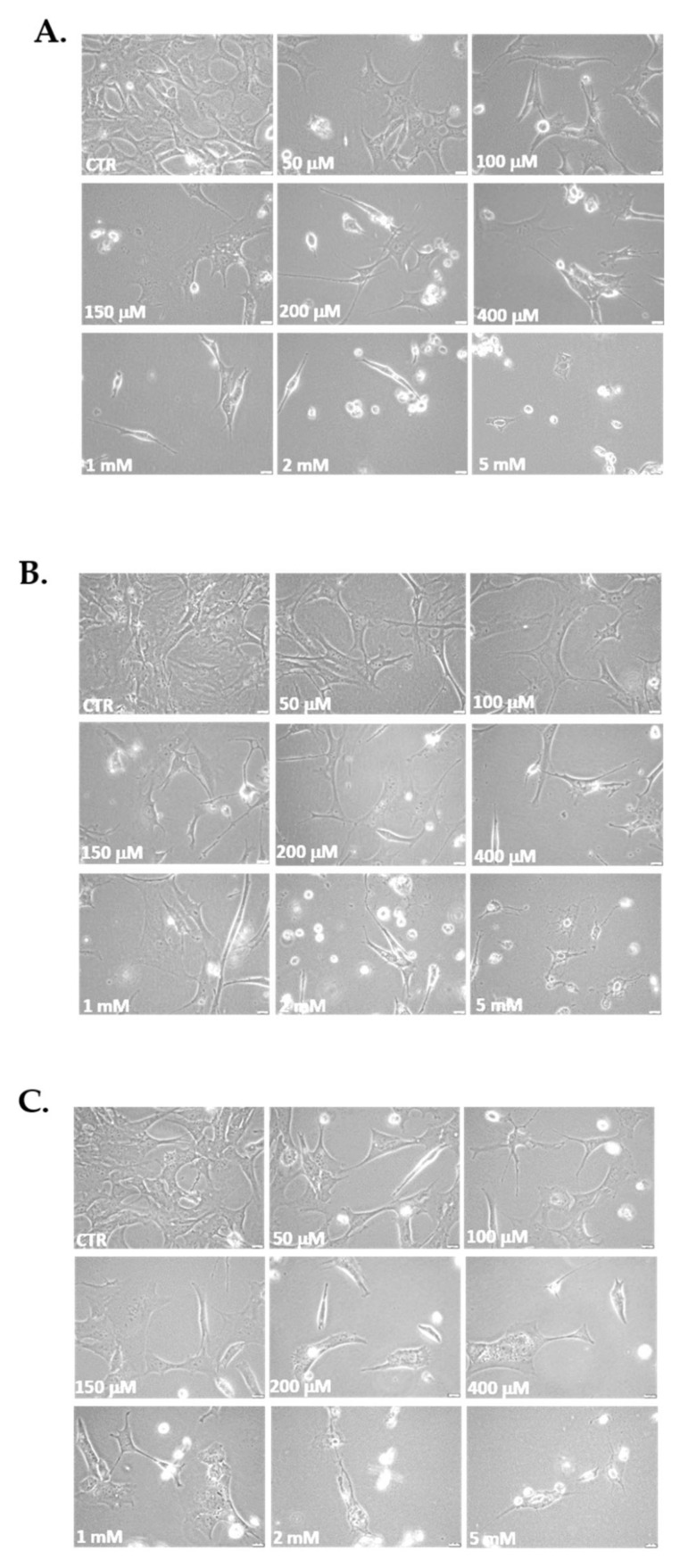
Morphological changes and capacity of cells to reseed post-H_2_O_2_ treatments. Representative images of morphological changes of cells after replated at 72 h post-treatment: control (CTR) cells and cells treated with H_2_O_2_ 50, 100, 150, 200 and 400 μM and 1, 2 and 5 mM for 1 h. (**A**) HEI-OC1 cell line; (**B**) OC-k3 cell line; and (**C**) SV-k1 cell line. Cells were observed (at 40×) with the Olympus CKX41 microscope and the coupled cellSens Entry imaging system. Scale bars: 100 μm.

**Figure 4 antioxidants-10-01497-f004:**
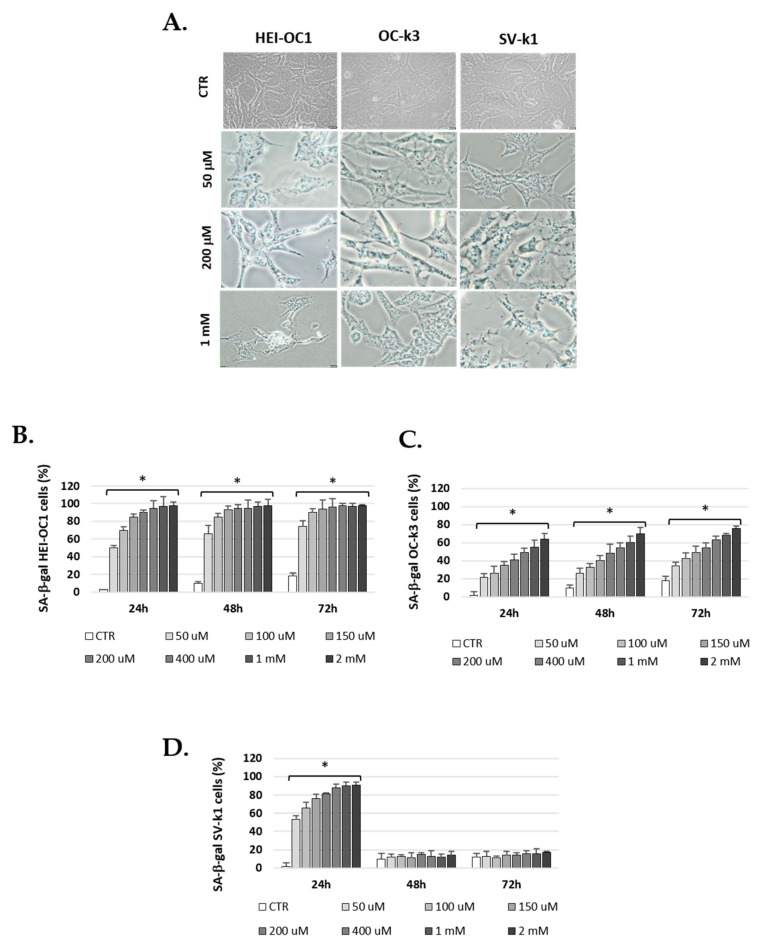
H_2_O_2_ induces early senescence in cells. (**A**) Representative imaging of SA-β-gal staining corresponding to HEI-OC1, OC-k3, and SV-k1 cells, treated with H_2_O_2_ 50, 200 mM and 1 mM for 1 h and 24 h post-treatment; (**B**–**D**) Percentage of SA-β-gal-stained auditory cells treated with H_2_O_2_ 50, 100, 150, 200, 400 μM and 1 and 2 mM for 1 h and 24, 48 and 72 h post-treatment. Data represent the means ± SD (*n* = 3). * *p* < 0.05 vs. CTR group.

**Figure 5 antioxidants-10-01497-f005:**
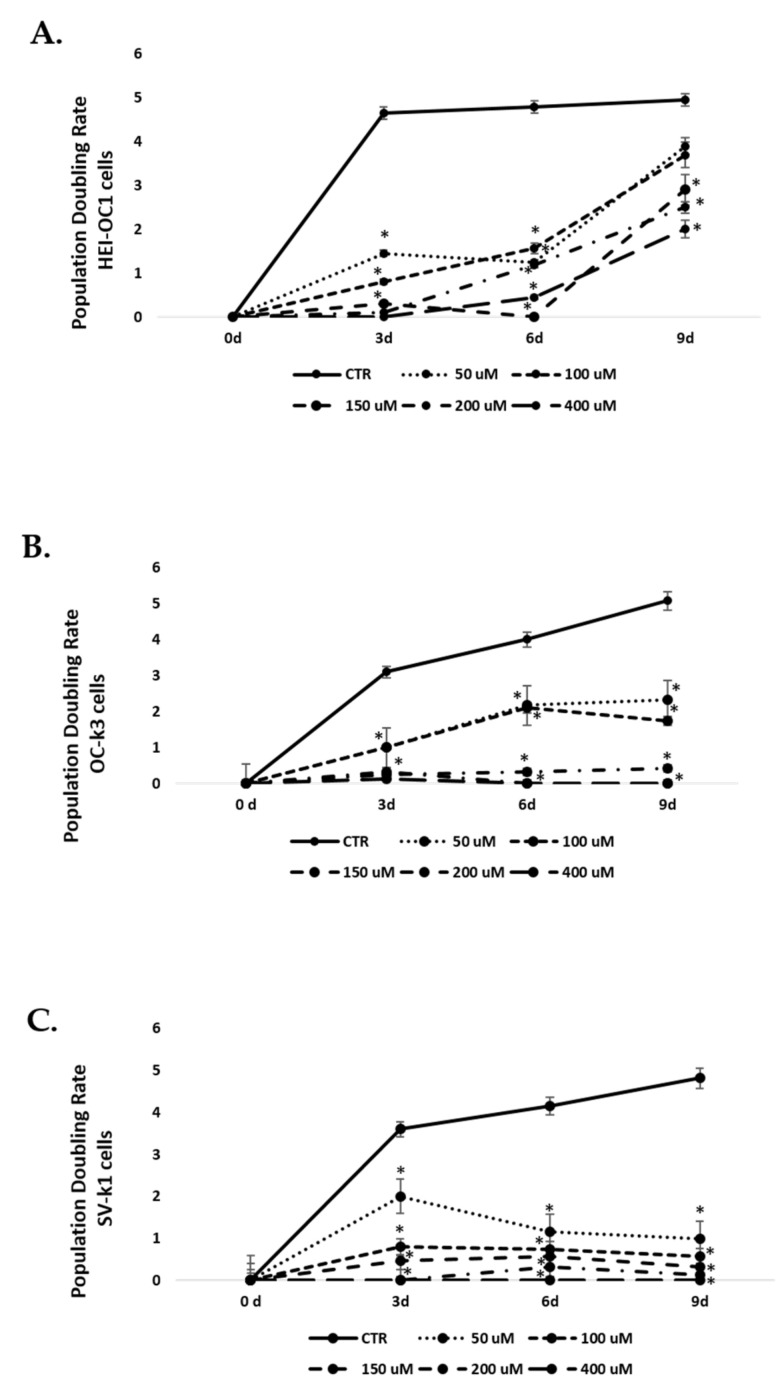
Decrease in population doubling rate in cells post-H_2_O_2_ treatment. Population doubling experiments were performed post-H_2_O_2_ treatments with 50, 100, 150, 200 and 400 μM H_2_O_2_ for 1 h. (**A**) HEI-OC1 cell line; (**B**) OC-k3 cell line; and (**C**) SV-k1 cell line. All values are means ± SD (*n* = 3). **p* < 0.05 vs. CTR group.

**Figure 6 antioxidants-10-01497-f006:**
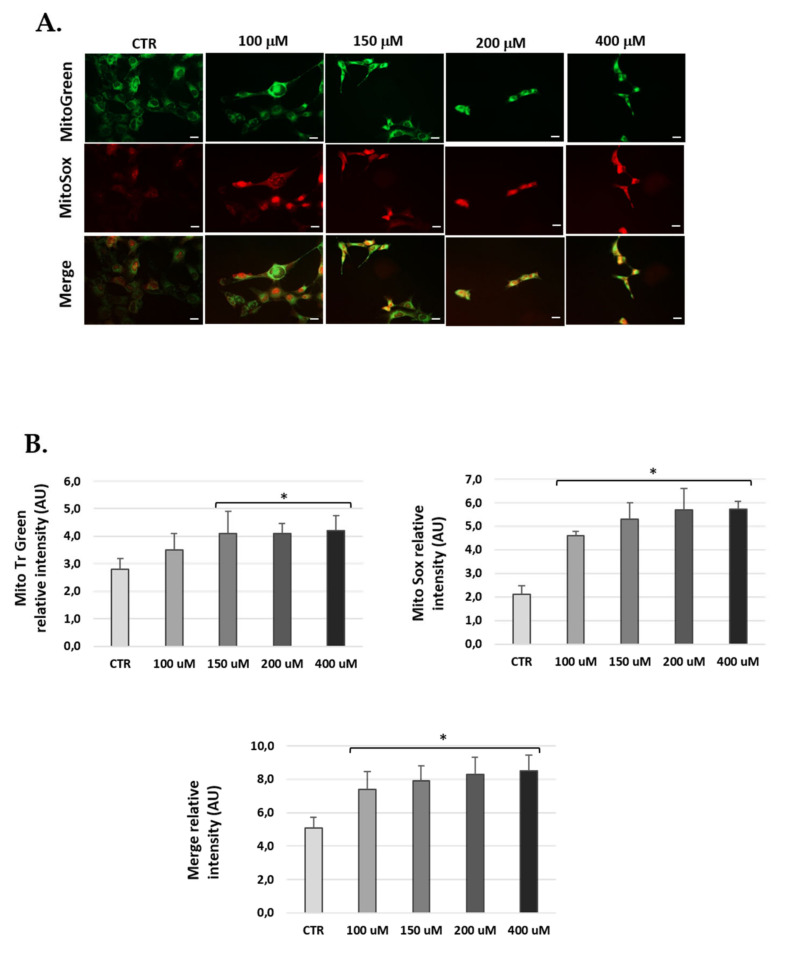
Mitochondria-derived ROS production in HEI-OC1 cells post-H_2_O_2_ treatment. (**A**) Representative fluorescence micrographs of MitoTracker Green, MitoSOX red and merge (yellow) in HEI-OC1 cells treated with 100, 150, 200 and 400 μM H_2_O_2_ for 1 h and 24 h post-treatment. (**B**) The graphs show the fluorescence intensity obtained by the ImageJ program. Data represent the mean ± SD of arbitrary units (AU) (*n* = 3). * *p* < 0.05 vs. CTR (*n* = 3).

**Figure 7 antioxidants-10-01497-f007:**
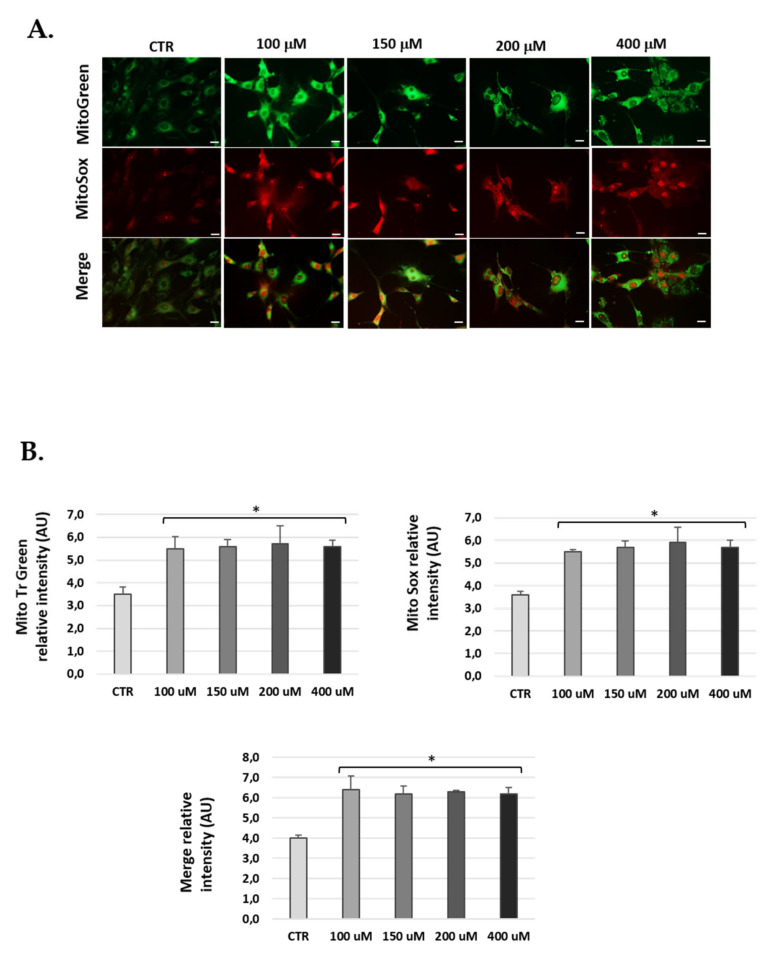
Mitochondria-derived ROS production in OC-k3 cells post-H_2_O_2_ treatment. (**A**) Representative fluorescence micrographs of MitoTracker Green, MitoSOX red and merge (yellow) in OC-k3 cell treated with 100, 150, 200 and 400 μM H_2_O_2_ for 1 h and 24 h post-treatment. (**B**) The graphs show the fluorescence intensity obtained by the ImageJ program. Data represent the mean ± SD of arbitrary units (AU) (*n* = 3). * *p* < 0.05 vs. CTR (*n* = 3).

**Figure 8 antioxidants-10-01497-f008:**
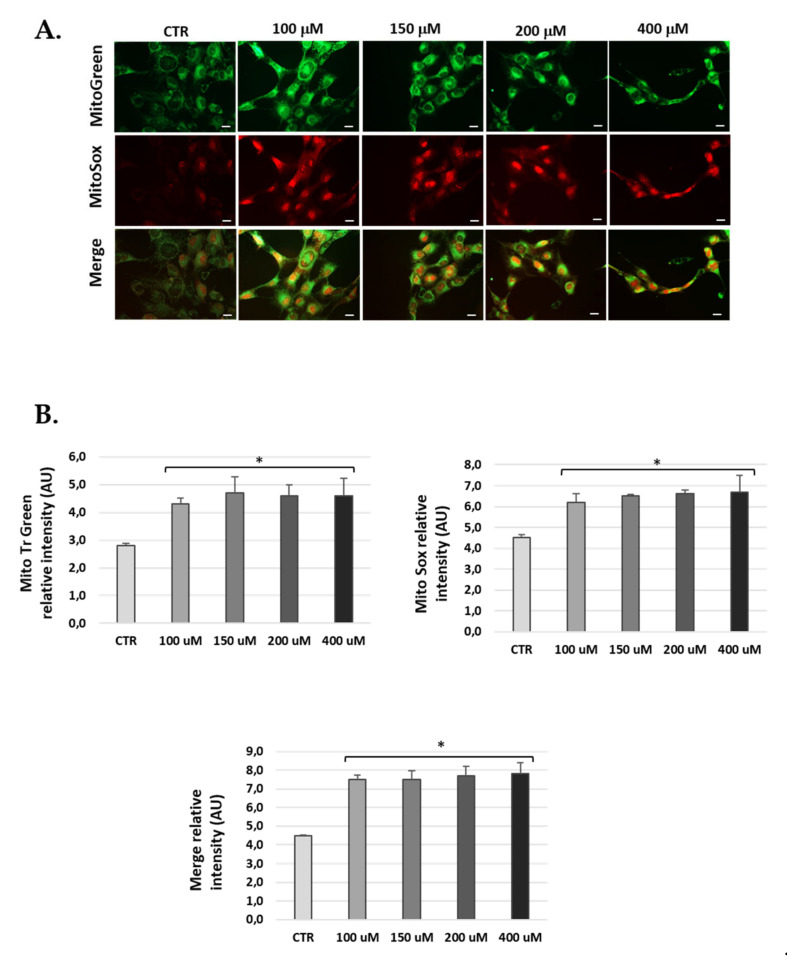
Mitochondria-derived ROS production in SV-k1 cells post-H_2_O_2_ treatment. (**A**) Representative fluorescence micrographs of MitoTracker Green, MitoSOX red and merge (yellow) in SV-k1 cell treated with 100, 150, 200 and 400 μM H_2_O_2_ for 1 h and 24 h post-treatment. (**B**) The graphs show the fluorescence intensity obtained by the ImageJ program. Data represent the mean ± SD of arbitrary units (AU) (*n* = 3). * *p* < 0.05 vs. CTR (*n* = 3).

**Figure 9 antioxidants-10-01497-f009:**
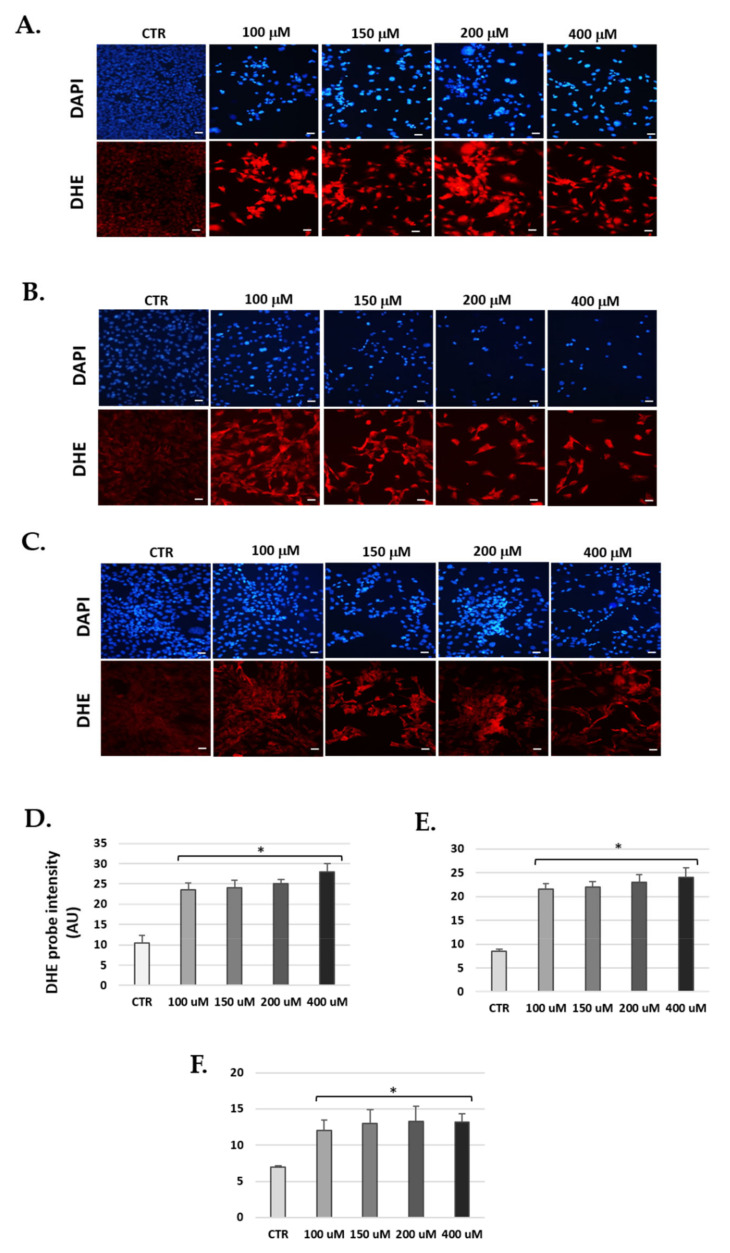
H_2_O_2_ increases the production of ROS in cells. Representative images of fluorescence micrographs of (**A**) HEI-OC1, (**B**) OC-k3 and (**C**) SV-k1 cells (×40). Scale bar 100 μm. (*n* = 4). ROS detection with probe DHE (red fluorescence), and DAPI dye (a marker of cell nuclei with blue fluorescence) in cell cultures at 24 h post-H_2_O_2_ treatment. Experimental groups: CTR (control); 100, 150, 200 and 400 μM for 1 h. (**D**–**F**) Quantification of fluorescence intensity was performed by ImageJ program. (**D**) HEI-OC1, (**E**) OC-k3 and (**F**) SV-k1 cells. Graphs represent the mean ± SD of arbitrary units (AU). * *p* < 0.05 vs. CTR. (*n* = 4).

**Figure 10 antioxidants-10-01497-f010:**
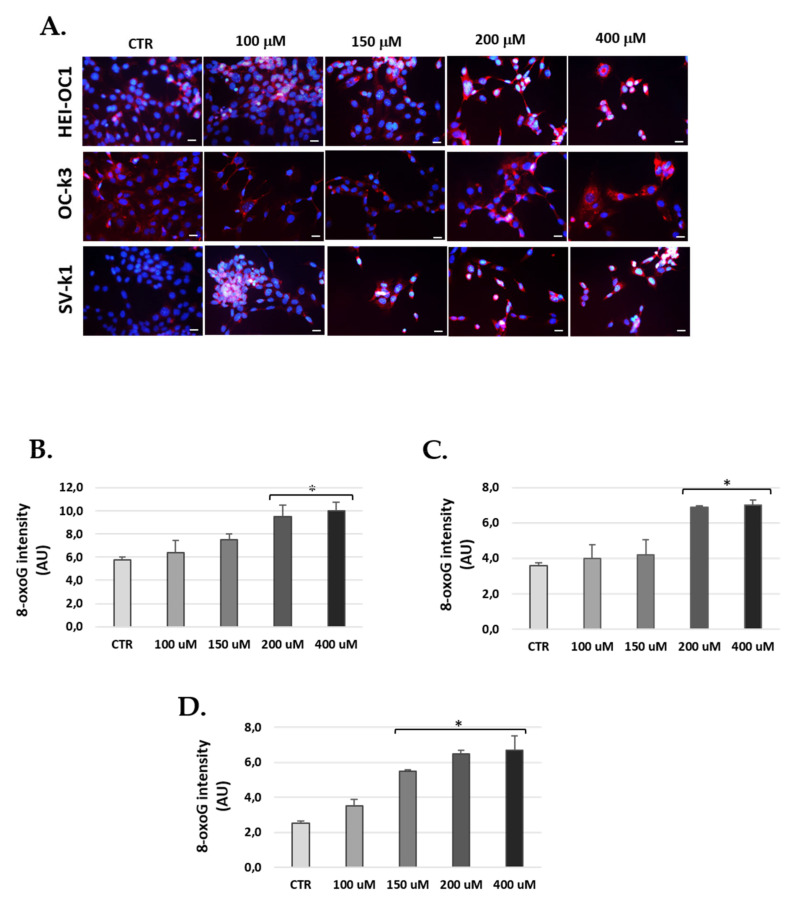
H_2_O_2_ increase production of 8-oxoG in cells. (**A**) Representative images of immunostaining of 8-oxoG in cell cultures treated with 100, 150, 200 and 400 μM H_2_O_2_ for 1 h and 24 h post-H_2_O_2_ treatment. DAPI: blue fluorescence (nuclei marker) and 8-oxoG immunoreactivity: red fluorescence. Scale bar: 100 μm. (**B**–**D**) Bar graphs show the immunofluorescence intensity changes calculated by ImageJ. (**B**) HEI-OC1, (**C**) OC-k3 and (**D**) SV-k1 cells. The values represent the mean ± SD (*n* = 3). * *p* < 0.05 vs. CTR (*n* = 4).

**Figure 11 antioxidants-10-01497-f011:**
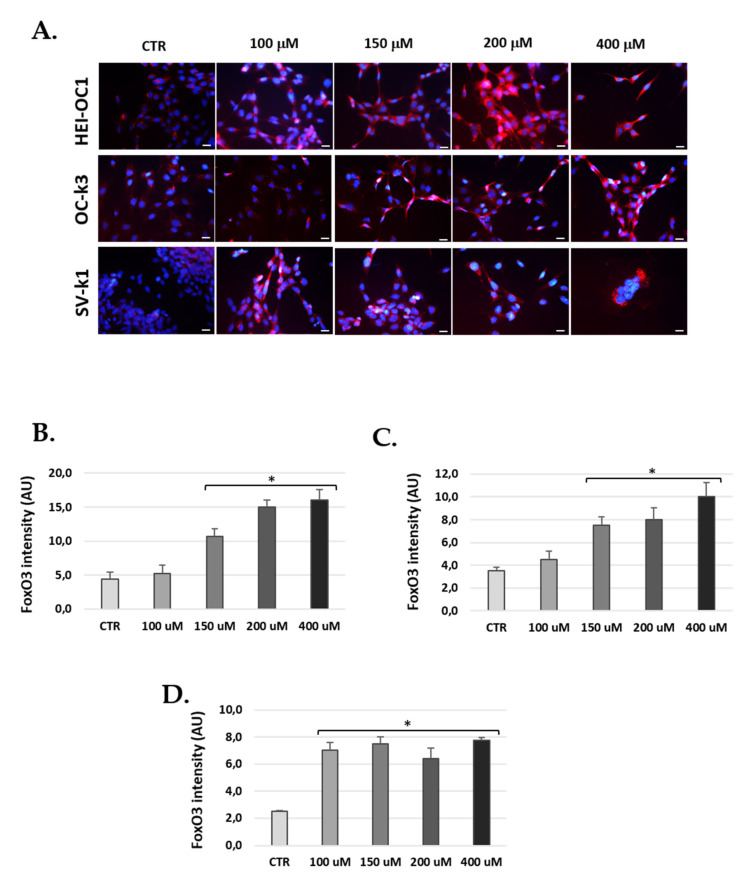
H_2_O_2_ increased production of Foxo3 in cells. (**A**) Representative images of immunostaining of Foxo3 in cells cultures treated with 100, 150, 200 and 400 μM H_2_O_2_ for 1 h and 24 h post-H_2_O_2_ treatment. DAPI: blue fluorescence (nuclei marker) and Foxo3 immunoreactivity: red fluorescence. Scale bar: 100 μm. (**B**–**D**) Bar graphs show the immunoflu-orescence intensity changes calculated by ImageJ. (**B**) HEI-OC1, (**C**) OC-k3 and (**D**) SV-k1 cells. The values represent the mean ± SD (*n* = 3). * *p* < 0.05 vs. CTR (*n* = 4).

## Data Availability

Data is contained within the article.

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
