# Peer review of "Role of Oxidative Stress in the Senescence Pattern of Auditory Cells in Age-Related Hearing Loss"

_antioxidants, 2021, doi:10.3390/antiox10091497_

Round 1
Reviewer 1 Report
The introduction about ARHL is too brief and restrictive. Add at least two sentences.
Concerning the result part, the authors do not often specify the reason for the experiment and it is difficult to see the relevant conclusions. It would be better to highlight the obtained results.
Please check abbreviation all along the manuscript. For example, ARHL, line 30, line 42 and line 78.
The abbreviation SASP was used for secretory senescence-associated phenotype. Wouldn't it be SSAP ?
Line 101 : CO2, not CO2.
Lines 241, 247, 340, 403 and 477: H2O2, not H2O2.
The term "hair cells" was used for the first time line 456: indicate cochlear hair cells (HC) and remove (HC) line 471.
line 479: write aging, not ag-ing.
lines 192 and 200: Figure 1 instead of Figure 2 and Figure 3
Add A, B and C on Figure 1 beside each graph
Reviewer 2 Report
Authors analyze effects of oxidative stress (induced by H2O2) in three well characterized auditory culture cell lines (SV-k1, HEI-OC1 and OC-k3). This design has been developed to check the proof of concept that ARHL can by triggered by ROS generation in the inner ear. In general terms the manuscript is well written, and methodology is adequate to reach the objectives.
My only concern is that there is no evidence in the paper showing that the auditory phenotype is preserved, at least in controls, at short and long term. One specific marker (prestin, synaptic receptors, potassium channels or any other specific markers) will be enough to be able to properly call these cells as -auditory -. Alternatively, authors can change the term auditory for the acronyms of the cell lines.
Most than probably reader will appreciate at the beginning of the Material and Method section a general detailed description of the time line of the cultures. So, 2.1 and 2.2 can be together to fully explained experimental design.
158 Please include details about the fixation (%, etc.)
163/171 please provide a detailed descriptions of the method (Image J) used to count cells.
Figure 1 have difficulties for a quick interpretation of data. 1 - Asterisk and line in each moment means -all conditions are significant between them-?. They are not significant differences with controls? The time evolution along 72 h in the bottom of the graphs may by confusing for the readers. Please provide a more descriptive legend of the figure.
